# Forecasting the Status of Municipal Waste in Smart Bins Using Deep Learning

**DOI:** 10.3390/ijerph192416798

**Published:** 2022-12-14

**Authors:** Sabbir Ahmed, Sameera Mubarak, Jia Tina Du, Santoso Wibowo

**Affiliations:** 1UniSA STEM, University of South Australia, Adelaide, SA 5001, Australia; 2School of Engineering and Technology, Central Queensland University, Melbourne, VIC 3000, Australia

**Keywords:** waste prediction, municipal, deep learning, time series, waste management

## Abstract

The immense growth of the population generates a polluted environment that must be managed to ensure environmental sustainability, versatility and efficiency in our everyday lives. Particularly, the municipality is unable to cope with the increase in garbage, and many urban areas are becoming increasingly difficult to manage. The advancement of technology allows researchers to transmit data from municipal bins using smart IoT (Internet of Things) devices. These bin data can contribute to a compelling analysis of waste management instead of depending on the historical dataset. Thus, this study proposes forecasting models comprising of 1D CNN (Convolutional Neural Networks) long short-term memory (LSTM), gated recurrent units (GRU) and bidirectional long short-term memory (Bi-LSTM) for time series prediction of public bins. The execution of the models is evaluated by Mean Absolute Error (MAE), Mean Absolute Percentage Error (MAPE), Coefficient determination (R^2^) and Root Mean Squared Error (RMSE). For different numbers of epochs, hidden layers, dense layers, and different units in hidden layers, the RSME values measured for 1D CNN, LSTM, GRU and Bi-LSTM models are 1.12, 1.57, 1.69 and 1.54, respectively. The best MAPE value is 1.855, which is found for the LSTM model. Therefore, our findings indicate that LSTM can be used for bin emptiness or fullness prediction for improved planning and management due to its proven resilience and increased forecast accuracy.

## 1. Introduction

Waste management is one of the greatest concerns anywhere in the world [1]. The random process of the rapid growth of population in urban areas makes the situation a more dynamic and challenging task for municipalities [2,3]. Therefore, evaluating the variation and causes of municipal waste generation becomes a challenging task to apply management strategies [4]. Proper and efficient forecasting is very much critical to generating an efficient system infrastructure for smart waste management. Moreover, optimized forecasting is essential to evade the spectacle of inadequacies of waste management and aid policymakers in generating advanced measures to minimize complications [5]. Many barriers exist to the implementation of digital waste management systems, including a lack of policymakers’ knowledge as well as the deficiency of standards and strategic rules [6]. Hence, numerous analysts have greatly contributed to exploring the impact of diverse components on the waste management era and various models to anticipate waste generation [7,8].

Municipal waste generation is noted to be composite and does not have any direct relationship between causes and effects concerning multiple aspects [9]. Broadly, two major tasks accomplished by the waste management system are waste collection and processing. In this research, we have considered the waste collection process which includes bins, vacant schedules, routing paths and pick-up locations. Improper waste collection can cause serious problems for the city and its citizens [10]. For example, overflow of bins can make the environment dirty and unhealthy, especially in public places, which is hazardous to citizens’ health. Therefore, a proper vacant schedule, appropriate bin placement and uncertainty prediction are huge challenges [11,12,13].

To overcome the problems and challenges of waste management, researchers are working from different perspectives such as municipal solid waste, organic waste, industry or chemical waste, medical waste and of waste recycling [8,14,15,16,17]. The architecture and algorithms of an assortment of deep learning strategies have been compared and evaluated to execute waste management activities [8]. With the utilization of deep-learning models, raspberry pi and camera module, a real-time smart waste monitoring system is designed for the photographic identification and categorization of waste [18]. Combining IoT and machine learning, Mookkaiah et al. [19] proposed a model that recognizes the category of waste and classifies them as biodegradable or non-biodegradable to gather them in individual waste containers. Several studies performed the e-waste classification using a machine-learning model to collect and remove the waste [20,21]. Fasano et al. [9] have applied the deep-learning models with permutation methods for dealing with the waste classification problem. The study determines the factors which are the most influencing variable, estimating the waste management independent variables and forecasting the effects from those factors. The studies include the different deep-learning models being utilized to develop the smart waste management system. 

On the other hand, several researchers reported that various models and hybrid techniques have been used to predict municipal waste used for quantity estimation, segregation and recycling of solid waste [11]. An artificial neural network (ANN) is widely used in a complex system, which demonstrates its performance in processing nonlinear data [22]. Deep learning is a popular method incorporating advanced technologies including big data and breakthroughs in training methods [23,24]. The Recurrent Neural Network (RNN) and Convolution Neural Network (CNN) are typical deep-learning algorithms [5]. CNNs are broadly used to identify objects from image and image segmentation [25]. RNNs are a variation of the routine feed-forward artificial neural systems that can employ consecutive information and be prepared to hold the information around the past. CNNs and RNNs have been applied to big data processing in different sectors in complex problems such as waste management.

This paper aims to analyze and forecast future trends in waste generation from the dataset generated by sensors located in smart bins. This study explores the potential deep-learning methods applied to the smart bin dataset to predict waste generation. The key idea behind these deep-learning models is to acquire a trustworthy estimation model so that the waste management authority can rely on the historical data generated from the bins. Specifically, the paper attempts to answer the following research questions: (1) How can deep learning models be used to forecast waste generation for helping decision-makers in waste-collection operations? and (2) What is the best model to forecast future waste generation? 

## 2. Related Study

In many digital waste administration perspectives, big data analytics can be used to incorporate machine learning and artificial intelligence. Gupta et al. [26] surveyed machine-learning models for garbage collection, sorting and reusing of garbage. Several projects are being conducted to help the government authorities in using machine learning or other information analytics procedures to deal with waste administration issues [6]. 

From an ecological perspective, deep-learning algorithms have been considered by several researchers to estimate waste generation [27,28,29,30,31]. LSTM, GRU and Bi-LSTM models are RNN, which are special sorts of artificial neural networks adjusted to work for time arrangement information or information that includes sequences. To overcome complex problems, researchers have developed many deep-learning methods including LSTM, which has been commonly pragmatic in time-series investigation [29]. The LSTM unit uses the three-gate architecture where it can help to determine whether the impact on municipal waste generation is transient and what the length of such an impact is in terms of executing training. Few studies have applied LSTM neural systems in the time-series investigation of municipal waste to estimate or resolve their temporal variety [30]. The BiLSTM model is used for capturing different suitable data from spatial time series data and estimating the resource consumption in a cloud-based data center [28]. Zhang et al. [31] outlined the development of a novel dynamic forecasting model based on the GRU with a time series investigation for displacement prediction. Few types of research used LSTM, GRU and Bi-LSTM for sequential data analysis such as demand forecasting [32], and many other diverse fields including waste management prediction.

As summarised in Table 1, several researchers have applied different models for estimating waste at different periods, for instance, weekly, seasonal or yearly [20,24,25]. Niu et al. [29] have considered two-year data starting from January 2018 to December 2019 to predict the MSW amount in Suzhou, China, using the LSTM model. Considering the data-driven features, some studies have applied ANN models. Chhay et al. [33] have contemplated eight socio-economic factors in China from the statistical yearbook 2000–2016 and the outcome attained in terms of MAE, MAPE, RMSE and R^2^. Abbasi and El Hanandeh [34] considered the ANN model for an 18-year period of data from Logan City, Australia. The potentiality of predicting the waste amount in Shanghai China using deep-learning methods has been quantified using the prediction accuracy measured by Lin et al. [5], and the result indicates the correlation coefficient of attention, 1D CNN and LSTM. Vu et al. [35] have considered 36 scenarios with ANN model revealed the changes in travel distance compared to the non-modified composition. Moreover, this study found that dual-compartment trucks save travel distance, slightly reducing emissions but increasing the collection time compared to single-compartment trucks. Due to waste sorting, collection, scheduling and disposal procedures, more measures are still required to enhance the performance and efficiency of waste management systems.

Waste administration mainly depends on the waste data produced from different bins placed in various places. Hence, the challenge from the managerial perspective is to have the proper vacant schedules, appropriate bin placement and uncertainty prediction. Thus, this paper focuses on four models such as 1D CNN, LSTM, GRU and Bi-LSTM for estimating future waste generation.

## 3. Materials and Methods

To achieve the target of this study, we trained these four models (1D CNN, LSTM, GRU and Bi-LSTM) to forecast the waste generation from public bin data produced by sensors. To evaluate the performance of the models, MAE, MAPE, R^2^ and RMSE performance indices have been measured. Then, considering the advantages and disadvantages in evaluating the performance and comparing between the models, five phases were accomplished by: (1) collecting the data generated from the public bins, (2) investigating and envisioning the data, (3) training the four categories of deep-learning models, (4) testing the models, and (5) mining and comparing the outcomes. Figure 1 shows the methodology of this study and illustrates the process starting from data collection to the performance evaluation of the models.

### 3.1. Deep Learning Models

For different applications in various sectors, the deep-learning procedures have shown the significant contributions of development reside in the literature. This section describes the basic principle of four deep learning models that will be used for waste generation prediction using smart bin data, namely 1D CNN, LSTM, GRU and Bi-LSTM.

#### 3.1.1. LSTM Model

Due to the gradient difficulties in deep neural or recurrent networks, the LSTM architecture was introduced by Hochreiter and Schmidhuber to overcome the problems in training long-term dependencies [36,37]. The LSTM design comprises a set of repetitively associated sub-networks denoted as memory blocks. The thought behind the memory block is to preserve its state over time and control the data stream through non-linear gating units. Figure 2 shows the structure of an LSTM block, which includes the doors, the input flag x(t), the yield h(t), the enactment capacities and peephole associations [38]. The input block and all the gates are recurrently connected with the output block. 

The computing of the LSTM network is the mapping of the input sequence, i.e., $X=\left(X_{1},X_{2},\cdots \cdots ,X_{n}\right)$ and output sequence, i.e., $Y=\left(Y_{1},Y_{2},\cdots \cdots ,Y_{n}\right)$. LSTM is calculated utilizing the following equations: (1)$$f_{t}=\sigma \left(W_{f}\;\cdot \;\left[h_{t-1},\;x_{t}\right]+b_{f}\right)$$
(2)$$i_{t}=\sigma \left(W_{f}\cdot \left[h_{t-1},\;x_{t}\right]+b_{i}\right)$$
(3)$$O_{t}=\sigma \left(W_{0}\;\cdot \;\left[h_{t-1},\;x_{t}\right]+b_{0}\right)$$
(4)$$C_{t}=f_{t}*C_{t-1}+i_{t}*tanh\left(W_{c}\;\cdot \;\left[h_{t-1},\;x_{t}\right]+b_{c}\right)$$
(5)$$O_{t}=\sigma \left(W_{0}\;\cdot \;\left[h_{t-1},\;x_{t}\right]+b_{0}\right)$$
(6)$$h_{t}=o_{t}*tabh\left(C_{t}\right)$$

In Equations (1)–(3), the activation of the input, output, and forget gates are denoted by $i_{t}$, $o_{t}$ and $f_{t},$ respectively. Here, the bias and weight variables symbolized by $b_{f}$, $b_{i}$, $b_{0}$, $b_{c}$, $W_{f}$, $W_{0}$, $W_{c}$ and $h_{t-1}$ signify the prior hidden layer units. After processing Equation (4), $C_{t\;}$ is converted to a current memory cell, which is the activation vector. Moreover, the sigmoid function is represented in Equation (5) and denoted by $\sigma \left({}^{\circ}\right)$. Additionally, Equation (6) shows the hidden layer outputs according to the element-wise multiplication, and adding nonlinearity on top is utilized by tanh and sigmoid functions, which are demonstrated in Equations (1)–(6).

#### 3.1.2. GRU Model

GRU is a model that chooses a new kind of hidden unit by utilizing the architecture of the LSTM unit [38,39]. GRU is the straightforward variation of LSTM that includes two gates: an input gate into a single update gate and a forget gate. The update gate consisting of the inputs from GRU has no extra memory cell to keep data which subsequently can regulate the data inside the unit [40]. The GRU architecture is illustrated in Figure 3.

GRU has exemplified its adequacy in an assortment of applications requiring consecutive or temporal information [41]. The transition capacities in hidden layers of the GRU cell are controlled according to Equations (7)–(10).
(7)$$z_{t}=\sigma \left(W^{z}x_{t}+V^{z}h_{t-1}+b^{z}\right)$$
(8)$$r_{t}=\sigma \left(W^{r}x_{t}+V^{r}h_{t-1}+b^{r}\right)$$
(9)$${\bar{h}}_{t}=tanh\left(W^{c}x_{t}+V^{c}\left(r_{t}\;\cdot \;h_{t-1}\right)\right)$$
(10)$$h_{t}=\left(1-z_{t}\right)\cdot \;h_{t-1}+z_{t}\;\cdot \;{\bar{h}}_{t}$$

Here, a gate controller, *z* in Equation (7), controls both input and forget gates. The forget gate is open and the input gate is closed while the *z* value is 0. However, the forget gate is closed, and the input gate is open when *z* is 0. At each step, the previous $\left(t-1\right)$ memory is saved, and the input of the time step is cleared. Moreover, the ${\bar{h}}_{t}$ executes the same function as in recurrent unit and in terms of $\;h_{t}$ where time *t* signifies the linear exclamation between the current ${\bar{h}}_{t}$ and previous $h_{t-1}$ activation inside the GRU unit stated in Equations (9) and (10).

#### 3.1.3. Bidirectional LSTM Model

Due to its design, an LSTM network can execute only the forward passes on consecutive information, which eventually implies the unidirectional model of data processing [42]. An instinctive way to mitigate this confinement is to utilize a clone copy of the LSTM arrangement but in the opposite order proposed by Schuster and Paliwal [43]. Hence, combining the LSTM forward and backward networks, a Bidirectional LSTM (BiLSTM) is made, which can be utilized to show conditions bidirectionally. Figure 4 demonstrates the structure of the BiLSTM network, which is an arrangement of handling sequence processing that comprises two LSTMs networks: one taking the input in a forward layer and the other in reverse order in a backward layer. The forward layer is accompanied by the inputs coming from the input layer, and the backward layer generates the outcomes in the output layer.

The bidirectional RNN encompasses two different hidden layers with similar output but in reverse orientation. Following the construction of BiLSTM, the output layer includes the previous and future information. For BiLSTM, the input sequence $X=\left(X_{1},X_{2},\cdots \cdots ,X_{n}\right)$ is designed in forward network such as $\vec{h_{t}}=\left(\vec{h_{1}},\;\vec{h_{1}},\cdots \cdots ,\;\vec{h_{n}}\right)$ and backward network as $\overset{\leftarrow }{h_{t}}=\left(\overset{\leftarrow }{h_{1\;}},\;\overset{\leftarrow }{h_{1}},\;\cdots \cdots ,\overset{\leftarrow }{h_{n}}\right)$. The final output generated from the sequence as the output of prediction vectors is $Y_{t}=\left[\cdots y_{t-1},y_{t},\;y_{t+1}\cdots \right]$. The final output cell is formed by both $\vec{h_{t}}$ and $\overset{\leftarrow }{h_{t}}$. Therefore, individual $y_{t\;}$ is calculated by combining both directions denoted in Equation (11):(11)$$y_{t}=\delta \left(\vec{h_{t}},\overset{\leftarrow }{h_{t}}\right)$$

### 3.2. Performance Indices

Four performance measures such as MAE, MAPE, R^2^ and RMSE are applied to estimate the performance of the proposed models. Here, C signifies the actual value, $\tilde{C}$ for estimated value and $\bar{C}$ is the average of the actual values. For the best model, MAE values should be equal to zero [44].
(12)$$\mathrm{MAE}=\frac{1}{M}\overset{M}{\underset{i=1}{\sum }}\left|C-\tilde{C}\right|$$

RMSE is specified in Equation (13) [44] as
(13)$$\mathrm{RMSE}=\sqrt{\frac{1}{M}\overset{M}{\underset{i=1}{\sum }}{\left(C-\tilde{C}\right)}^{2}}$$

MAPE is specified in Equation (14) [44] as
(14)$$\mathrm{MAPE}=\frac{1}{M}\overset{M}{\underset{i=1}{\sum }}\frac{\left|C-\tilde{C}\right|}{\left|C\right|}$$

R^2^ is specified in Equation (15) [44] as
(15)$$R^{2}=\frac{\sum_{i=1}^{M}{\left(\tilde{C}-\bar{C}\right)}^{2}}{\sum_{i=1}^{M}{\left(C-\tilde{C}\right)}^{2}}$$

### 3.3. Data Description

We have used data obtained from public bins from a particular council located in Wyndham city, Melbourne, Australia. These data were obtained from the data portal: https://data.gov.au [45]. It contains the json file, and the data are stored according to the json tags. The recorded data were collected from the council starting from July 2018 to May 2021. Table 2 illustrates the specification of the dataset for waste generation. The collected dataset contains the bin data generated from the sensors, and data from a total of thirty-two bins are stored every day. However, the bins are placed in various public places within the Wyndham city council area. Different attributes are used to identify the suitable bin to be vacated using the filling-level detection. A detailed description of each attribute is presented in Table 2. It can be observed that the first attribute defining it is a collection point, the second and third attributes storing the geographical position for a bin. An even number has been used for setting the criteria and tracking the status (latest empty/fullness) of a bin used by four, nine and ten number attributes. The reason attributes define the verbal status generated from the numeric values. The serial number is the tracking number for a particular bin and description containing the string describing the details of a particular bin. The position is an attribute that defines where the bin has been placed, and finally the timestamp attribute which contains the date information. First, the data is pre-processed for missing values before it is applied to deep-learning models.

### 3.4. Data Exploration

Understanding the data trends and behaviors is essential and valuable while dealing with data analysis. Moreover, this observation evaluates the meaningful identification of the fact produced by the data [46]. Figure 5 demonstrates the seasonal decomposition of the bin status (empty/fullness) from the one-month waste generation data. It shows the great similarity between the residual trend and the observed components. We can observe interesting information extracted from the data, considering the trend and seasonality. Based on the plot in Figure 5, the trend in the data seems to be high overall. Moreover, there is apparent regularity within the information, causing the bin status to fluctuate by 0.5 over the whole period. Furthermore, the residuals are obtaining periods of high variableness throughout the time series, and the randomness in the data will be smooth through the moving average (MA), which can aid the model.

## 4. Results and Discussion

This section presents the results on the deep-learning models that are trained-tested. Finally, the model performances are evaluated through statistical indices. For sequential forecasting, the 1D CNN, LSTM, GRU and BiLSTM methods are chosen. The waste generation dataset is divided into training samples of 80% consisting of 292 days and a test sample of 73 days. The performance of the deep learning models is calculated by executing MAE and RMSE standard statistical error measurement indices. The total experiment and the simulations are executed in Google-Colab [46] and encoded with Python deep learning API Keras [47].

The dataset comprises the feature of the bin status (fullness threshold) and timestamp. The unscaled data take longer for the convergence activities. The MinMaxScaler (scales the data to a specific value range) estimator will fit the training data set in the case of normalizing the training and test data sets, and the same estimator will be utilized to transform both training and test data sets. The shape of the uniqueness of disseminating data is preserved by MinMaxScaler. Furthermore, it also maintains valuable information about outliers and scales the data to a limited range of values. Table 3 presents the proposed models’ parameters with their assessments of 1D CNN, LSTM, GRU and BiLSTM is represented in.

For all the four models, the optimizer is Adam, batch size is 70 and epochs are 20. It is worth mentioning here that all the aforementioned parameters are modified through trial and error. The next step is to calculate the train and test error calculation for the 1D CNN, LSTM, GRU and BiLSTM models. The error values are illustrated in Table 4 while considering performance measure indices MAE, MAPE, R^2^ and RMSE. The smallest value of MAE and RMSE is considered the best model. Following the criteria, we found that LSTM is the best compared to the others. The RMSE value for the LSTM model is the lowest. Therefore, LSTM is more efficient at forecasting long-term dependencies in comparison to 1D CNN, GRU and BiLSTM. 

Figure 6 displays the plots train loss and valid loss with various epochs applied to the models to predict waste generation. With the increase of epochs, the loss grows higher for both training and valid datasets. For 1D CNN, the trend of loss grows after 10 epochs and also in 6 epochs, reaching 0.15 and 0.15, respectively. The trend of loss in the train loss increases with the increase in the epochs for LSTM and BiLSTM reaching 0.01 and 0.10, respectively at 20 epochs. However, for GRU, the train loss decreases while the epochs increase reaching 0.16 at 20 epochs. The train loss and test loss almost maintain constancy with the values 0.02 and 0.08 at epoch 4, which demonstrates that the GRU model has achieved the convergence state. Therefore, the GRU has a better prediction performance for waste generation.

Figure 7 displays the differences between the actual and predicted results. In a few incidents, the predicted outcome diverges from the actual outcome, while executing the simulation using models.

The correctness of the models is tabulated in Table 4. The RMSE and R^2^ of the LSTM model are the lowest with the values 1.579 and 0.925, respectively. Therefore, LSTM is more capable of forecasting waste generation trends than 1D CNN, GRU and BiLSTM are, with a very low variance between others. The visual description of the predicted and actual outcomes of the training dataset for four models is illustrated in Figure 7. From the representation of the results and the advantages and disadvantages of the 1D CNN, LSTM, GRU and BiLSTM models, the conclusion can be drawn that LSTM is suitable for the waste data to predict waste generation. The process of managerial decision-making will be more efficient with the forecasted results using the LSTM model.

The proposed models in this paper can be considered reliable and acceptable for waste estimation. Table 5 encapsulates the summary that compares the models in the literature and those in this study [5,29,30,31,32,33,34]. Niu et al. [29] have considered two-year data using the LSTM model and have obtained RMSE 940 and coefficient determination (R^2^) of 0.90. Chhay et al. [33] have contemplated eight socio-economic-factors in China from the statistical yearbook 2000–2016 and the outcome attained in terms of MAE, MAPE, RMSE and R^2^ are 228.53, 0.0143, 450.84, and 0.931, respectively. Abbasi and El Hanandeh [34] considered the ANFIS compared to ANN model for 18 years and achieved R^2^ (0.99–0.83), MAE (0.001–335.03), RMSE (0.002–498.43), and MAPE (3.39 × 10^−6^–0.07). Lin et al. [5] depict that the correlation coefficient of attention, one-dimension CNN, and LSTM are 78, 86.6 and 90, respectively, while considering the actual and prediction values. Based on the values 3.17, 1.85, 2.42, 7.95 for 1D CNN, LSTM, GRU, and Bi-LSTM from MAPES, the best forecasting model is LSTM compared to all other models. The prediction and the actual results are very close with RSME values measured for 1D CNN, LSTM, GRU and Bi-LSTM models at 1.12, 1.57, 1.69 and 1.54, respectively. These results show that LSTM performs better than other models in terms of predicting empty/fullness status of the bins within a year. 

Waste generation is predicted by different seasonal periods such as daily, weekly, monthly or yearly, and a few researchers have used the gross or total amount of waste data produced by the bins as a whole for a particular region [5,30,33,34]. However, this study has considered individual garbage bins located in public places and predictions are being made from the bin data generated from the sensors. As a result, the proposed models show more efficient forecasting than other existing models. Furthermore, compared to other models, LSTM performs better in predicting the empty/fullness status of the bins. This outcome confirms the best use of resources, dynamic deployment bins with a better vacant schedule. This model demonstrates the usefulness and reproducibility in different municipals. Moreover, this deep learning model can be applied to waste analysis and administration decision-making problem in various settings.

## 5. Conclusions

In this study, deep learning methods can be observed to have a noteworthy impact on technological advances, particularly in creating diverse time-series-based forecast models. This study aims to examine the predictive models while applying the data coming from endpoints such as sensors, rather than using the human-made historical dataset. The predictive analysis for waste generation revealed different behavior originating from different sources located within a council. 

Waste generation forecasting is the primary stage for waste management and waste disposal processes. The study has applied four types of deep-learning algorithms such as 1D CNN, LSTM, GRU and BiLSTM for predicting waste generation. To test the effectiveness of the four models, performance measure metrics are calculated. The result reveals that LSTM outperformed the other models with an MAE of 0.602, MAPE 1.855, R^2^ 0.925 and RMSE 1.579. In addition, 1D CNN, LSTM, GRU and BiLSTM have shown robustness and much-enhanced forecasting when assessed with actual numbers representing less estimate errors. The changes in the waste generation period have various significant values that could interrupt the waste collection, transportation and disposal process. Therefore, detailed knowledge of these changes revealed from the deep-learning method could immensely help waste management authorities during emergencies or for estimating a proper vacant schedule to ensure a better sustainable environment for the citizens and local council. The deep learning models are used for waste generation analysis using the dataset produced from the sensors located in the bin. For practical implications, these deep learning models can be modified and adopted by waste management authorities to acquire a reliable waste analytic system while relying on the data produced from the bins. 

In future work, we will be exploring different factors such as organizational, socioeconomic factors and public utility investment that might have an impact on waste generation. Another study can also be done to identify the influential factors through the in-depth study intervention for the development of a more optimized and cost-efficient waste management system.

## Figures and Tables

**Figure 1 ijerph-19-16798-f001:**
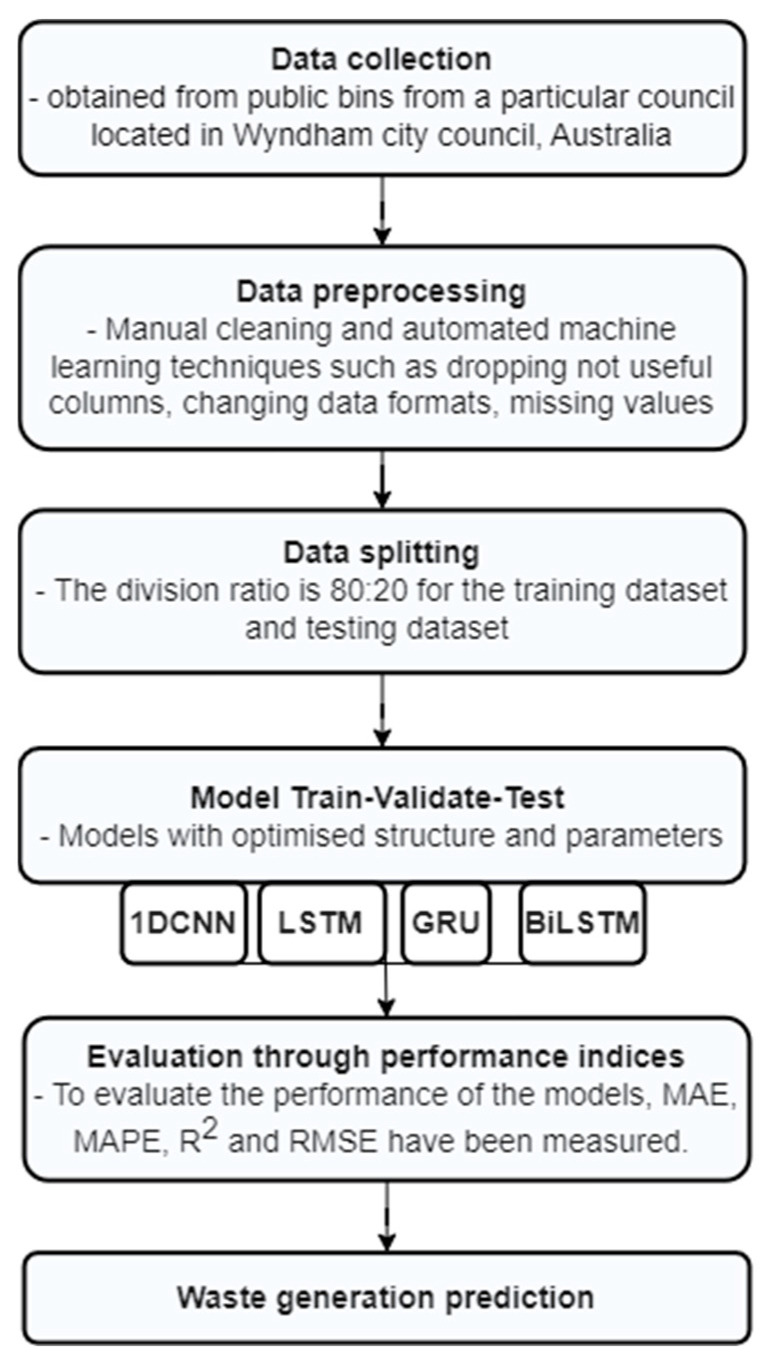
Methodology to deal with the data processing and selecting models.

**Figure 2 ijerph-19-16798-f002:**
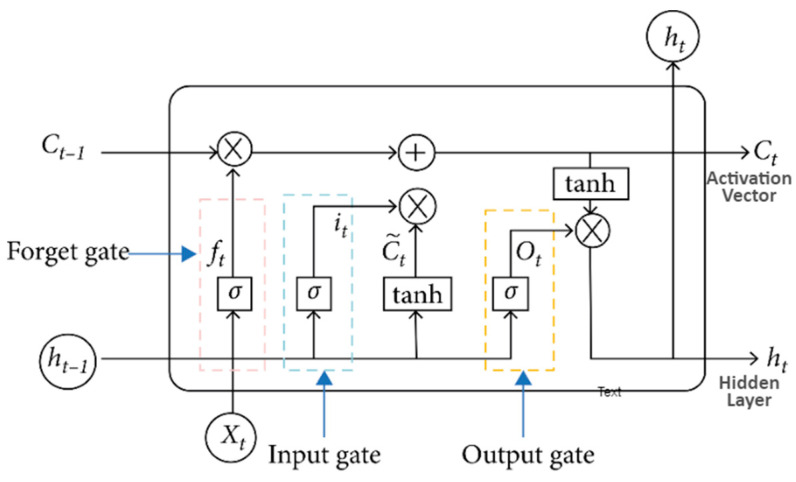
Architecture of LSTM.

**Figure 3 ijerph-19-16798-f003:**
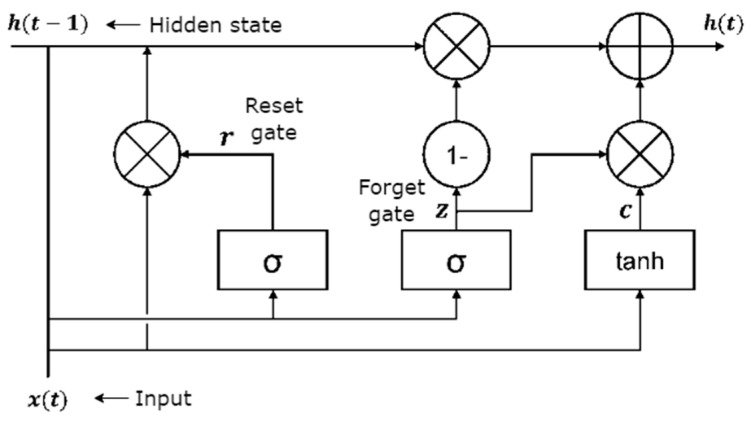
Architecture of GRU.

**Figure 4 ijerph-19-16798-f004:**
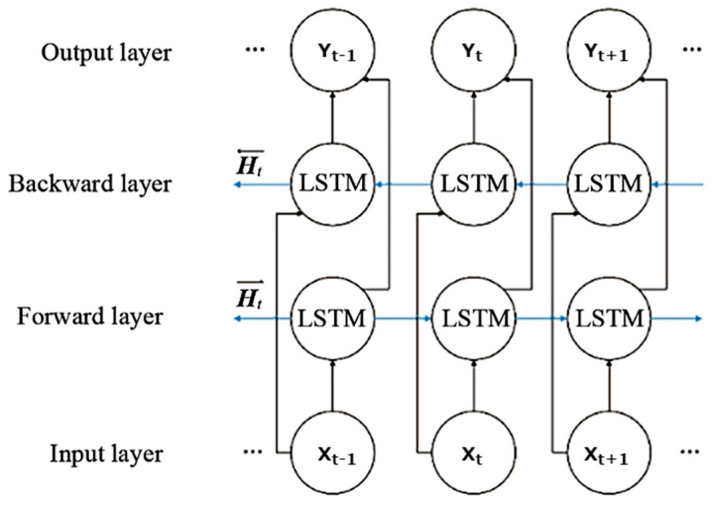
Architecture of Bi-LSTM.

**Figure 5 ijerph-19-16798-f005:**
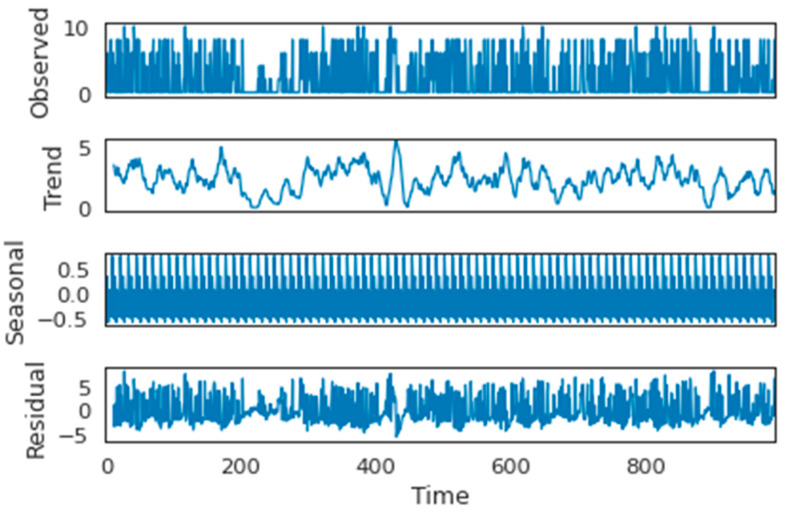
Simulated and decomposed components for the waste dataset.

**Figure 6 ijerph-19-16798-f006:**
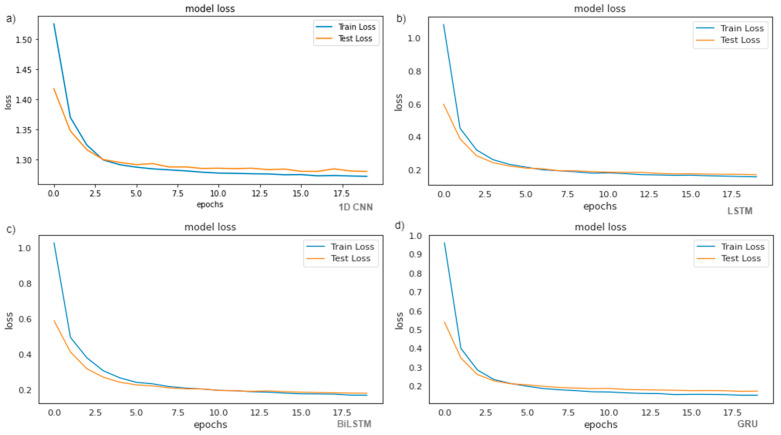
Prediction performances according to four models (loss vs. epochs): (**a**) Loss of the 1D CNN vs. different epochs; (**b**) Loss of the LSTM vs. different epochs; (**c**) Loss of the BiLSTM vs. different epochs; (**d**) Loss of the GRU vs. different epochs.

**Figure 7 ijerph-19-16798-f007:**
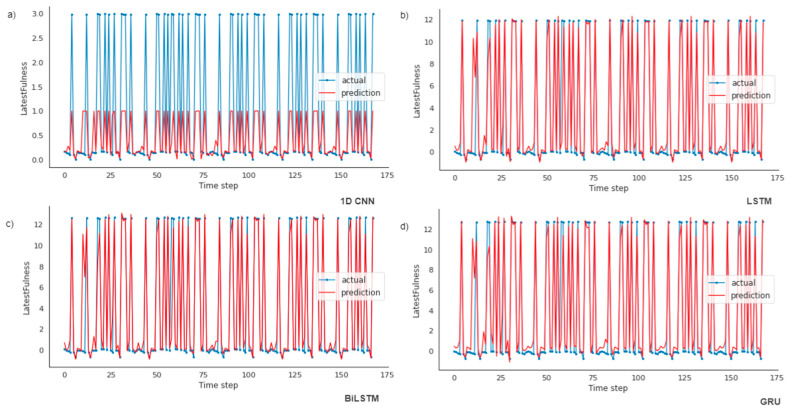
Actual and predicted bin fullness using four models: (**a**) Plot of actual vs. predicted values for 1D CNN model; (**b**) Plot of actual vs. predicted values for LSTM model; (**c**) Plot of actual vs. predicted values for BiLSTM model; (**d**) Plot of actual vs. predicted values for GRU model.

**Table 1 ijerph-19-16798-t001:** Application of deep–learning models in waste generation prediction.

Application	Region	Dataset	Findings	References
Waste generation using ANN	Logan City, Australia	Waste generation (July 1996 to June 2014)	18-year period historical dataset did not use the smart bin data	Abbasi and El Hanandeh [34]
Predicting waste amount using LSTM	Suzhou, China	730 data (January 2018 to December 2019)	Time series dataset from a historical record for five different districts. The data is as a whole amount the waste for a district on a particular day.	Niu et al. [29]
Predicting waste amount using Attention, 1D CNN, LSTM and 1D CLA	Shanghai, China	January 1990 to December 2018	Considered 24 socioeconomic factors and the historical dataset did not use the smart bin data	Lin et al. [5]
Forecasting waste generation ANN, Linear regression	Shanghai, China	Date from 2000 to 2016	Contemplated eight socio-economic-factors with the historical dataset not using the smart bin data	Chhay et al. [33]
Predicting waste generation	Melbourne, Australia	Date from 2010 to 2020	Multiple socioeconomic factors data are available for conducting further research including the smart bin	Watson and Ryan [12,13]
ANN	Austin, USA	Weekly collected garbage (2004–2018)	Predicting the garbage generation using weekly amount of collected garbage	Vu et al. [35]

**Table 2 ijerph-19-16798-t002:** Detail description of the collected data set.

S. No.	Attribute	Attribute Information
1	Type	Points
2	Coordinates0	Geographical position Latitude
3	Coordinates1	Geographical position Longitude
4	LatestFullness	Numeric 0 to 10
5	Reason	Fullness and Not_Ready
6	SerialNumber	Numeric number
7	Description	Details about the points
8	Position	Centre
9	AgeThreshold	Numeric 0 to 10
10	FullnessThreshold	Numeric 6 or 8
11	Timestamp	Date

**Table 3 ijerph-19-16798-t003:** Characteristics of the proposed deep learning models.

Layer	Output Shape	Parameters	Parameter Details		
			Total Parameter	Trainable Parameter	Non-Trainable Parameter
1D CNN	(None, 1, 200)	6200	6401	6401	0
GlobalMaxPooling1D	(None, 200)	0			
Dense	(None, 1)	201			
LSTM	(None, 100)	52,400	52,501	52,501	0
Dropout	(None, 100)	0			
Dense	(None, 1)	101			
GRU	(None, 100)	39,600	39,701	39,701	0
Dropout	(None, 100)	0			
Dense	(None, 1)	101			
BiLSTM	(None, 200)	104,800	105,001	105,001	0
Dropout	(None, 200)	0			
Dense	(None, 1)	201			

**Table 4 ijerph-19-16798-t004:** Comparison among deep-learning models in terms of different error measures.

Model	MAE		MAPE		RMSE		R^2^	
	Train	Test	Train	Test	Train	Test	Train	Test
1D CNN	0.667	0.677	3.170	3.678	1.128	1.132	0.274	0.269
LSTM	0.602	0.705	1.855	2.198	1.579	1.798	0.925	0.903
GRU	0.698	0.811	2.427	2.787	1.694	1.937	0.921	0.897
BiLSTM	0.638	0.747	7.951	8.911	1.543	1.774	0.925	0.901

**Table 5 ijerph-19-16798-t005:** A comparison between previous work and the models proposed in this study.

Purpose	Models	Results	References
Waste generation	ANFIS	MAE: 0.001, MAPE: 3.39 × 10^−6^ RMSE: 0.002, R^2^: 0.99	Abbasi and El Hanandeh [34]
	ANN	MAE: 335.03, MAPE: 0.07 RMSE: 498.43, R^2^: 0.83	
Estimating waste amount	LSTM	MAPE: 63.66, RMSE: 659.58, R^2^: 0.96	Niu et al. [29]
Predicting waste amount	LSTM	R^2^: 0.90	Lin et al. [5]
	Attention	R^2^: 0.78	
	CNN	R^2^: 0.86	
Predicting waste generation	ANN	MAE: 228.53, MAPE: 0.0143 RMSE: 450.84, R^2^: 0.931	Chhay et al. [33]
Estimating waste generation	1D CNN	MAE: 0.667, MAPE: 3.170,	The proposed models
		RMSE: 1.128, R^2^: 0.274	
	LSTM	MAE: 0.602, MAPE: 1.855,	
		RMSE: 1.579, R^2^: 0.925	
	GRU	MAE: 0.698, MAPE: 2.427,	
		RMSE: 1.694, R^2^: 0.921	
	BiLSTM	MAE: 0.638, MAPE: 7.951,	
		RMSE: 1.543, R^2^: 0.925	

## Data Availability

Data used in this study are available in publicly accessible repositories can be access through the links included in Section 3.3.

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
