# Peer review of "Forecasting the Status of Municipal Waste in Smart Bins Using Deep Learning"

_ijerph, 2022, doi:10.3390/ijerph192416798_

Round 1
Reviewer 1 Report
Please see attached file

Reviewer 2 Report
see attached

Reviewer 3 Report
The manuscript is well-written and shares novel findings. However, the reviewer suggests the following comments for the improvement of the presented work.
INTRODUCTION
Include more latest references, and more recent studies report on a similar topic in 2022. Include at least a few more references.
RELATED STUDY
Can you further widen the comparison to at least one more country, USA, UK, Japan---- in Table 1
MATERIALS AND METHODS
Which computer programs are used and why? clearly mention in this section.
RESULTS AND DISCUSSIONS
Improve the quality of figure 7
further, elaborate on the comparison presented in table 5. How one can use it for research or practical application.
CONCLUSIONS
Conclusions are very general, please elaborate with reference to the main outcomes of the study.
